# Microbial C and N Metabolism Alterations Based on Soil Metagenome and Different Shrub Invasion Stages in Sanjiang Plain Wetlands

**DOI:** 10.3390/microorganisms12081648

**Published:** 2024-08-12

**Authors:** Rongtao Zhang, Shenzheng Wang, Haixiu Zhong, Xiaoyu Fu, Lin Li, Li Wang, Yingnan Liu

**Affiliations:** 1State Key Laboratory of Urban Water Resource and Environment, Harbin Institute of Technology, Harbin 150090, China; zhangrongtao14@163.com; 2Institution of Nature and Ecology, Heilongjiang Academy of Sciences, Harbin 150040, China; zhx971030@163.com (H.Z.); 18646583130@163.com (X.F.); 3Heilongjiang Provincial Key Laboratory of Ecological Restoration and Resource Utilization for Cold Region, School of Life Sciences, Heilongjiang University, Harbin 150040, China; wangshenzheng2000@163.com; 4Key Laboratory of Forest Ecology and Forestry Ecological Engineering, Heilongjiang Ecology Institute, Harbin 150081, China; lilin_1002@163.com

**Keywords:** Sanjiang Plain, shrub expansion, soil microorganisms, metagenomes

## Abstract

Shrub invasion affects plant growth and soil physicochemical properties, resulting in soil microbiota metabolic pathway changes. However, little is known about the shrub expansion intensity of microbial metabolic pathway processes. In this study, we used metagenome sequencing technology to investigate changes in soil microbial C and N metabolic pathways and community structures, along with different shrub invasion intensities, in the Sanjiang Plain wetlands. Different shrub invasion intensities significantly affected the soil microbial composition (β diversity), with no significant effect on the α diversity compared to CK. AN, pH, and TP were the major factors influencing the microbial community’s structures. Compared to CK, the shrub expansion intensity did not significantly affect C fixation and central metabolism but significantly reduced methanogenesis, which involves the CO_2_-to-methane transition that occurs in methane metabolism, and denitrification, the nitrite to nitric oxide (nirK or nirS) transition that occurs in N metabolism. This study provides an in-depth understanding of the biogeochemical cycles of wetland ecosystems in cold northern regions undergoing shrub invasion.

## 1. Introduction

As an important type of wetland ecosystem, marsh wetlands have a unique ecosystem structure, complex and diverse habitats, and rich biodiversity, occupying an important position in the biogeochemical cycle of ecosystems [1,2]. However, marsh wetlands are being seriously degraded due to global warming and high-intensity human activities [3,4]. The decrease in wetland groundwater levels is particularly pronounced due to the increased frequency and intensity of activities such as agricultural reclamation and anthropogenic drainage [5], resulting in flood-intolerant shrub species invading marsh wetlands [6,7], with inevitable and significant impacts on their plants, soils, and C and N cycling processes. Soil microorganisms are the predominant mediators of C and N cycling in wetlands at the soil–plant–animal interface. As previous studies mainly investigated the effects of shrub expansion on soil microbial biomass, composition, and diversity [8,9,10], those on the soil microbial community’s function remain unclear.

Microorganisms are vital primary producers, decomposers, and drivers of biogeochemical cycles, playing a mediating role in organic matter decomposition and nutrient storage and turnover [11,12,13]. Shrub invasion affects soil’s microbial transformation by altering its physicochemical properties [14,15]. Li et al. [16] used phospholipid fatty acid technology to prove that shrub expansion significantly increased surface soil microbial biomass and the abundance of Gram-negative bacteria, clumped mycorrhizal fungi, and actinomycetes in both typical and desert grasslands. Meanwhile, Ding et al. [17] found that shrub expansion improved plant species’ richness and changed the assembly process and species richness of abundant and rare soil subcommunities, but not soil multifunctionality. Shrub expansion has also been shown to alter soil fungal diversity and its distributional characteristics, with gradient elevation in high-Arctic soils [18]. Zhou et al. [19] found that shrub expansion affected soil bacterial and fungal communities, mainly through increased phosphorus limitation. Although a lot of studies have proven that different shrub expansion intensities affect soil microbial community structure [20,21,22], few have focused on the effects on its functions, especially C and N cycling, with an overall lack of comprehensive discussions.

The Sanjiang Plain is the most complete, well-maintained natural marsh wetland in northeastern China. However, global climate change and anthropogenic disturbances have caused the degradation of these wetlands and a decline in their ecological functions, leading to a phenomenon of shrub expansion, with the obvious distribution of shrub and herbaceous patches in the gaps. Few studies have reported on the changes in soil microbial structure, function, and their controlling factors in marsh wetlands, or on different shrub expansion intensities. Therefore, in this study, we used metagenome technology to investigate the effects of shrub expansion intensity on the soil’s microbial structure, communities, and C and N cycling function in the marsh wetlands of the Sanjiang Plain, as well as its relationship to soil physicochemical properties, proposing two hypotheses: firstly, shrub expansion changes the soil’s physicochemical properties and microbial community structure; and, secondly, shrub expansion reduces the soil water content, further decreasing methanogenic pathway abundance.

## 2. Materials and Methods

### 2.1. Research Area Description

This study was performed at an experimental field station of the Heilongjiang Academy of Sciences, located in the Honghe National Nature Reserve (47°42′18″–47°52′07″ N, 133°34′38″–133°46′29″ E), Heilongjiang province, China (Figure 1). The area is approximately 1.1 × 10^4^ ha in size, mainly presenting *Deyeuxia angustifolia*-type wetlands [23]. It has a typical temperate humid/semi-humid monsoon climate, with an average annual temperature of 1.9 °C and average annual precipitation and evaporation values of 585 mm and 1166 mm, respectively [24]. The soil is classified as typical bleached stagnant soil and fibrous organic soil, and the dominant vegetation includes *Deyeuxia angustifolia*, *Glyceria spiculose*, *Carex lasiocarpa*, and *Carex pseudocuraica*.

### 2.2. Experimental Design

In July 2022, wetlands suffering from different intensities of invasion of *Spiraea salicifolia*, a typical representative shrub, were selected. The marsh shrub cover (a) was used to delineate four plot types by shrub expansion degree: a = 0, no expansion (CK); 0 < a ≤ 30%, mild expansion (SI); 30% < a ≤ 70%, moderate expansion (MI); and 70% < a ≤ 100%, severe expansion. Three samples were selected for each degree of shrub expansion, for a total of twelve different bog plots.

### 2.3. Soil Sample Collection and Analysis

In 2014, top-layer soils (about 0~20 cm) were collected from each plot using a 5 cm inner diameter auger. Then, 5~10 soil cores were randomly collected from each plot after litter removal. Each plot’s samples were mixed, stored in a Ziplock bag at 4 °C, and immediately transported to the laboratory. The soil samples were homogenized using a 2 mm mesh sieve and divided into two, storing one at −80 °C for microbial community analysis, while air-drying the other to determine the soil physicochemical properties.

### 2.4. Determination of Soil Physicochemical Properties

To determine the soil physicochemical properties, we followed the method described in our previous study [25]. Briefly, the soil pH was measured using a pH meter and a soil-to-water ratio of 1:2.5 *w*/*v*. The soil organic C and total N were measured using an elemental analyzer (Elementar, Langenselbold, Germany). The available N was examined via a continuous flow analysis (SAN++, Skalar Analytical, Breda, The Netherlands), while the total phosphorus was measured with a spectrophotometer. The available phosphorus was measured using a colorimetric method upon extraction with 0.5 M NaHCO_3_.

### 2.5. DNA Extraction, Metagenome Sequencing, and Data Processing

Genomic DNA was extracted using a commercial kit (ALFA-SEQ Advanced Soil DNA), according to the manufacturer’s instructions. The DNA was extracted three times for each treatment. The DNA’s integrity and purity were monitored on 1% agarose gels. The DNA’s concentration and purity were simultaneously measured using Qubit 3.0 and Nanodrop One (Thermo Fisher Scientific, Waltham, MA, USA). Sequencing libraries were generated using the ALFA-SEQ DNA Library Prep Kit, with their quality assessed via a Qubit 4.0 Fluorometer (Life Technologies, Grand Island, NY, USA) and a Qsep400 High-Throughput Nucleic Acid Protein Analysis System (Houze Biological Technology Co., Hangzhou, China). Finally, the libraries were sequenced on an Illumina NovaSeq 6000 platform, generating 150 bp paired-end reads [26].

Trimmomatic (v.0.36) was used to clean the raw data for subsequent analyses. MEGAHIT (Version v1.0.6) was used to assemble the clean data. Scaftigs (≥500 bp) assembled from both single and mixed samples were used to predict the open reading frame (ORF) with MetaGeneMark (Version 3.38). CD-HIT (Version: 4.7) was adopted to remove redundancy and obtain the unique initial gene catalog (UniGene), clustered by 95% identity and 90% coverage, with the longest sequence selected as the representative one. The clean data were compared with the gene catalog using BBMAP to calculate each gene’s abundance information in each sample.

BLASTP was used to compare the unigene sequences to gene sets from the NCBI non-redundant (NR) database. The final, aligned result of each sequence (e value ≤ 1 × 10^−10^) was selected, applying the lowest common ancestor (LCA) algorithm to the systematic classification of the MEGAN software (version 6.22.1, built 7 Mar 2022) to obtain the species annotation information of all sequences. The species composition and abundance information of each sample’s taxonomic hierarchy (kingdom, phylum, class, order, family, genus, and species) were obtained based on the LCA annotation results. Diamond software (v0.9.32.133) was used to blast the unigenes, alongside the Kyoto Encyclopedia of Genes and Genomes (KEGG) functional gene database, and the raw data were uploaded to the NCBI database, under number PRJNA1142744.

### 2.6. Statistical Analysis

A difference analysis of soil microbial communities was performed using the Shannon index and nonmetric multidimensional scaling (NMDS), as well as the analysis of similarity (ANOSIM) test. The relative abundance of different taxonomic levels was used to map the microbial community composition. In addition, a redundancy analysis (RDA) was used to assess the effect of soil physicochemical properties on the microbial community, determining its significance by a Monte Carlo permutation.

The diversity index was calculated as follows [27]:Shannon–Wiener (H): H = ∑(P_i_)(lnP_i_),
where P_i_ = N_i_/N, with N_i_ being the number of characteristic fatty acids in the treatment and n1, n2, and n indicating the number of individuals with the first, second, and nth characteristic fatty acid biomarker, respectively.

We conducted ANOSIM calculations with the R language (R3.2, New Zealand) anosim function, while the NMDS and RDA analyses and mapping used R3.2 (Vegan package).

## 3. Results

### 3.1. Soil Physicochemical Properties Changed with Shrub Expansion

Significant differences (*p* < 0.05) were observed in AN, TP, and AP with different shrub expansion intensities, with the exception of pH, SOC, and TN (Table 1). HI, MI, and SI significantly increased the AN and TP contents compared to CK (Table 1), while the AP content significantly increased under HI compared to CK (Table 1).

### 3.2. Comprehensive Characterization of Microbial Community Composition

The soil microbial Shannon index calculated based on single genes showed a significant increase under HI compared to CK (Figure 2A). The NMDS was used to show the β diversity of the soil microbial communities (Figure 2B), which was severely altered by different levels of shrub expansion (Figure 2, PERMANOVA: *p* = 0.01).

The dominant phyla in the microbial community were Acidobacteria (23.35%), Proteobacteria (17.93%), and Verrucomicrobia (11.88%), accounting for over 50% (Figure 3A), followed by Chloroflexi (7.68%), Actinobacteria (3.96%), Candidatus_Rokubacteria (2.31%), Gemmatimonadetes (1.53%), Planctomycetes (1.28%), and Nitrospirae (1.01%), whose abundance was >1%, but who were found in all soil samples. An increased shrub expansion intensity significantly reduced the relative abundance of Chloroflexi and Actinobacteria but increased that of Acidobacteria and Candidatus_Rokubacteria (Figure 3A).

*Bradyrhizobium* (1.62%) was the most abundant genus in all the soil samples (Figure 3B). The top 15 genera in terms of relative abundance were *Bradyrhizobium* (3.21%), *Candidatus_Sulfopaludibacter* (0.96%), *Candidatus_Sulfotelmatobacter* (0.79%), *Anaeromyxobacter* (0.45%), *Candidatus_Udaeobacter* (0.29%), *Usitatibacter* (0.28%), *Rhodoplanes* (0.25%), *Ktedonobacter* (0.24%), *Reyranella* (0.23%), *Trebonia* (0.23%), *Streptomyces* (0.22%), *Edaphobacter* (0.19%), *Pseudolabrys* (0.17%), and *Candidatus_Koribacter* (0.14%). An increased shrub expansion intensity significantly increased the relative abundances of *Bradyrhizobium*, *Rhodoplanes,* and *Reyranella* but reduced those of *Anaeromyxobacter* and *Ktedonobacter* (Figure 3B).

### 3.3. C and N Cycles under Different Shrub Expansion Intensities

Based on the KEGG results obtained via metagenomic sequencing, we analyzed the relative abundance of the corresponding pathways involved in C fixation and the central, methane, and N metabolism cycles (Figure 4). In the C fixation cycles, the dicarboxylate−hydroxybutyrate, Calvin−Benson−Bassham, and reductive cycles are the most important pathways (Figure 4A), while, in central metabolism, the TCA cycle, gluconeogenesis, and oxaloacetate to fructose−6P are the most important (Figure 4B). In methane metabolism, the methanogenesis pathways of acetate and CO_2_ to methane are the most important (Figure 4C), whereas, in N metabolism, the most important pathways are the denitrification of nitrite to nitric oxide (nirK or nirS), the dissimilatory nitrate reduction of nitrite to ammonia (nirBD or nrfAH), the dissimilatory nitrate reduction of nitrate to nitrite (narGHI or napAB), the nitrification of nitrite to nitrate (nxrAB), and the denitrification of nitric oxide to nitrous oxide (norBC) (Figure 4D). Increasing shrub expansion intensities did not significantly affect C fixation and central metabolism, but significantly reduced the methanogenesis of CO_2_ to methane in methane metabolism and the denitrification of nitrite to nitric oxide (nirK or nirS) in N metabolism (Figure 4).

### 3.4. Correlation Analysis of Microbial and Environmental Factors

A redundancy analysis was performed to identify the most important drivers behind the structural and compositional differences in the soil microbial communities. At the phylum level, the cumulative variations in the first and second dimension of the RDA plot were 65.4% and 23.7% (Figure 5A), respectively, while, at the genus level, they were 77.7% and 19% (Figure 5B). The soil pH, AN, and TP were the major factors explaining the soil microbial community’s composition at both levels. In addition, the combination of AN, AP, and TN was a key predictor of microbial community structure, as revealed by the RDA plots and further alignment tests. The three variables together explained 77% (phylum) and 82% (genus) of the variation in the microbial communities (Figure 6).

## 4. Discussion

### 4.1. Shrub Expansion Significantly Changed Soil Properties

Vegetation plays an important role in material and energy flow in wetland ecosystems, and there are spatial distribution differences between the characteristics and distribution of vegetation communities and wetland soils [28,29]. On the one hand, soil nutrients can be directly absorbed and assimilated by the vegetation’s root system and transformed into organic components such as nucleic acids, phospholipids, and chlorophyll, which, in turn, affect vegetation growth [30,31]. On the other hand, the vegetation can indirectly change the soil microenvironment (e.g., pH, water content, and organic carbon) through apoplastic and root secretion inputs, which would then negatively feedback-regulate the plant community’s structural composition and diversity [32,33]. Many studies have reported the shrub invasion effect in wetland soils, whereby soil nutrients (C, N, and P) are higher in the soils under shrub invasion compared to the original wetland soil [34,35]. In this study, wetlands under mild shrub invasion showed higher soil nutrient levels than the original wetland soil, which is consistent with previous studies [34,35]. Meanwhile, the soil nutrient levels in wetlands under moderate and severe shrub invasion did not change significantly compared to the CK conditions. It is possible for mild shrub invasion to lead to an increase in C inputs, affecting changes in apoplastic decomposition and microbial activity and, in turn, leading to an increase in soil organic C [36,37]. On the other hand, shrub expansion intensity increases lead to an increase in plant nutrient uptake from the soil, which, in turn, reduces the soil’s nutrient content. It has been found that the total phosphorus content of marsh soil increases with shrub expansion [15,38], which is consistent with our study results. In this study, the soil nutrient content was significantly altered as the degree of shrub expansion increased, suggesting that shrub expansion is an important factor regulating the process of nutrient recirculation in swamp soils.

### 4.2. Shrub Expansion Effects on Soil Microbial Community Structure

Previous studies have revealed that shrub expansion can significantly affect soil’s microbial diversity and composition [20,21]. In the present study, we found that HI significantly increased the α diversity of soil microbial communities, whereas there was no significant difference under SI and MI compared to CK. This suggests that the changes in the α diversity of soil microbial communities were related to the shrub expansion intensity, possibly due to the fact that mild shrub expansion did not alter the input of waste C sources and, thus, impact soil microbial diversity [22]. The dominant phyla were Acidobacteria, Proteobacteria, and Verrucomicrobia, which is consistent with the results of previous marsh wetland studies [23,39]. Consistent with these previous results, an increased shrub expansion intensity significantly increased the relative abundance of Acidobacteria, as members of this phylum are typically more adaptable to acidic soils and shrub expansion reduced the soil pH, leading to soil acidification [40]. Studies have shown that Actinobacteria have strong metabolic and repair functions and play an important role in organic matter turnover and C cycling [41]. Liu et al. [42] reported that an Acidobacterium abundance was significantly and positively correlated with the soil C content, differing from our results, possibly due to our use of metagenomics technology, which resulted in more precise experimental results.

Microbial communities are highly responsive to changes in the soil environment. Previous studies have emphasized that various environmental factors cause changes in the soil’s microbial structure and function [43,44]. In this study, we found that the nutrient contents of soil AN, TP, and SOC were significantly and positively correlated with the microbial community’s structure under shrub expansion. This could be attributed to soil nutrient changes due to shrub expansion, leading to further changes in the soil microbial community, validating our first hypothesis. The VPA further showed that most of the differences detected in the microbial community structure [77% and 82%] could be attributed to the AN, meaning that the environmental factors in the N content contributed the most to driving these changes. This is consistent with previous findings, revealing a N limitation in wetland microbial community shifts [45,46].

### 4.3. Shrub Expansion Effects on C and N Cycles

Microbial C sequestration is an important source of soil C accumulation [47,48]. In the Sanjiang Plain wetlands, six metabolic pathways of microbial C sequestration were noted, of which the dicarboxylate–hydroxybutyrate, Calvin–Benson–Bassham, and reductive cycles were the most important. Mild shrub expansion did not significantly affect the C sequestration pathways in the Sanjiang Plain wetlands, possibly due to the fact that short-term shrub expansion did not alter the quality or quantity of plant C inputs and, thus, did not significantly affect the C-sequestering microbial community and pathways [49]. In methane metabolism, the methanogenesis pathways of acetate to methane and CO_2_ to methane were the most important ones. Interestingly, shrub expansion significantly reduced the methanogenic pathway in these wetlands, mainly by reducing the transformation of CO_2_ to methane. This is consistent with previous studies [50] and mainly due to the decline in soil water content with shrub expansion, which reduces the anaerobic environment, inhibiting the abundance and activity of methanogenic bacteria and affecting the wetlands’ methanogenic pathway. This validates our second hypothesis.

Soil N is one of the key elements required by plants. Ammonification, ammonia oxidation, nitrification, and denitrification are the four main microbial processes associated with the supply, leaching, and transformation of N nutrients in soil systems [51,52]. In this study, the most important pathways were the denitrification of nitrite to nitric oxide (nirK or nirS), the dissimilatory nitrate reduction of nitrite to ammonia (nirBD or nrfAH), the dissimilatory nitrate reduction of nitrate to nitrite (narGHI or napAB), the nitrification of nitrite to nitrate (nxrAB), and the denitrification of nitric oxide to nitrous oxide (norBC). Shrub expansion significantly reduced the denitrification of nitrite to nitric oxide (nirK or nirS) in N metabolism. This is not consistent with the results of a previous study by Liang et al. [53], who found that shrub expansion increased the relative abundance of denitrification genes in desert ecosystems, as stronger nitroalkane decomposition and nitrification under shrubs provided NO_2_^−^ and NO_3_^−^, further facilitating the denitrification process. This difference in the findings may be due to the two different ecosystems, wetland and desert, and due to water limitation.

## 5. Conclusions

As expected, shrub expansion significantly altered the soil’s microbial composition and metabolic function but had no significant effect on its α diversity. AN, pH, and TP were the major factors influencing the structures and metabolic functions of soil microbial communities. Shrub expansion did not significantly affect C fixation and central metabolism but significantly reduced the methanogenesis of CO_2_ to methane in methane metabolism and the denitrification of nitrite to nitric oxide (nirK or nirS) in N metabolism. This study systematically revealed the effects of short-term shrub expansion on the soil microbial community’s structure and function. Future studies need to be conducted to record the response of soil microbial communities’ structure and function to long-term shrub expansion.

## Figures and Tables

**Figure 1 microorganisms-12-01648-f001:**
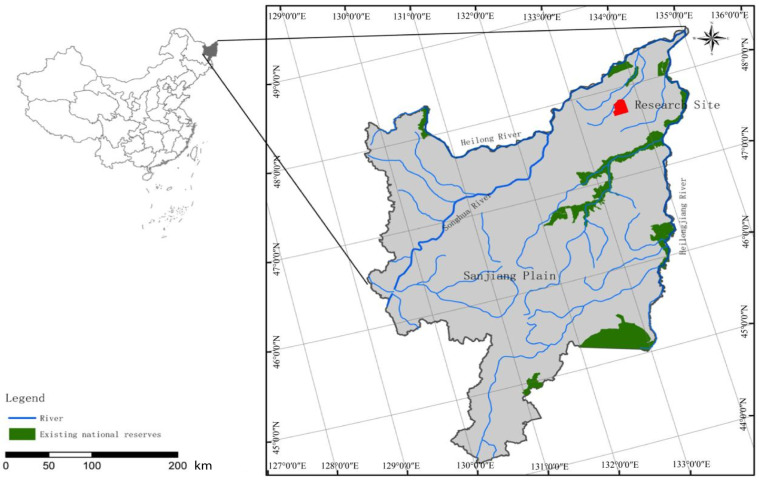
The location of this study.

**Figure 2 microorganisms-12-01648-f002:**
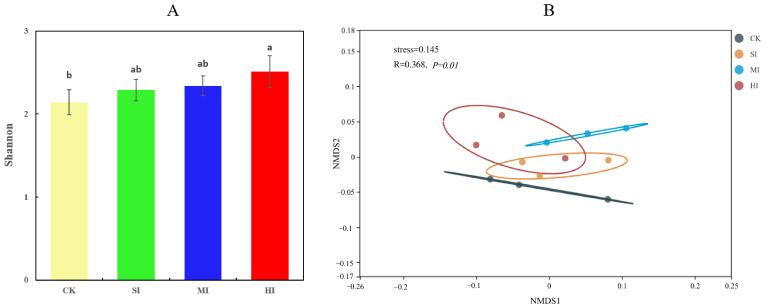
Alpha (**A**) and beta diversity (**B**) of soil microbial communities under different shrub invasion intensities. LSD post hoc comparison tests analyzed the significant differences; different letters represent significant differences (ab, *p* < 0.05). The error bars indicate the standard deviation of three replicates. CK, no expansion; SI, mild expansion; MI, moderate expansion; and HI, severe expansion.

**Figure 3 microorganisms-12-01648-f003:**
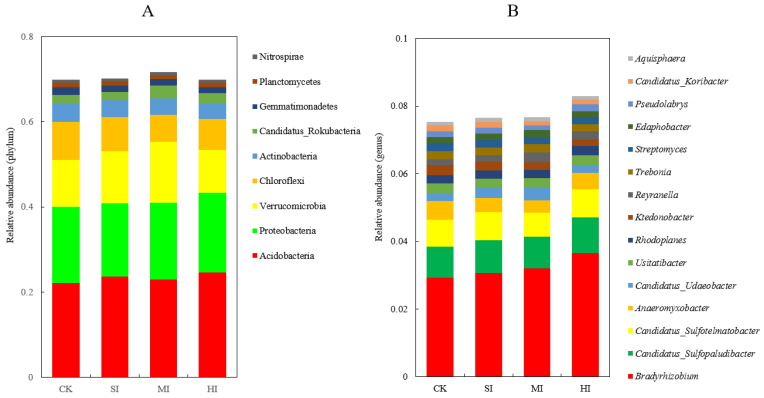
Relative abundance of microbes at the phylum (**A**) and genus level (**B**) under different shrub expansion intensities. CK, no expansion; SI, mild expansion; MI, moderate expansion; and HI, severe expansion. The percentages are the average content.

**Figure 4 microorganisms-12-01648-f004:**
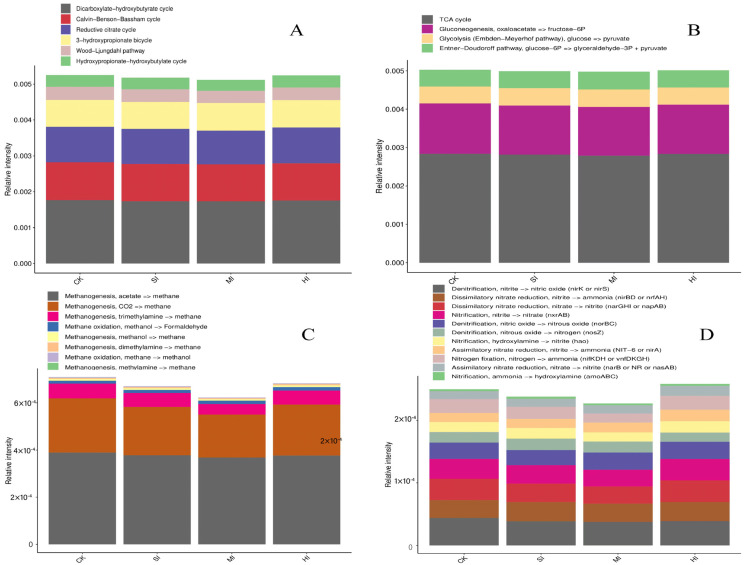
C fixation (**A**) and central (**B**), methane (**C**), and N metabolism (**D**) pathways under different shrub expansion intensities. CK, no expansion; SI, mild expansion; MI, moderate expansion; and HI, severe expansion. The percentages are the average content.

**Figure 5 microorganisms-12-01648-f005:**
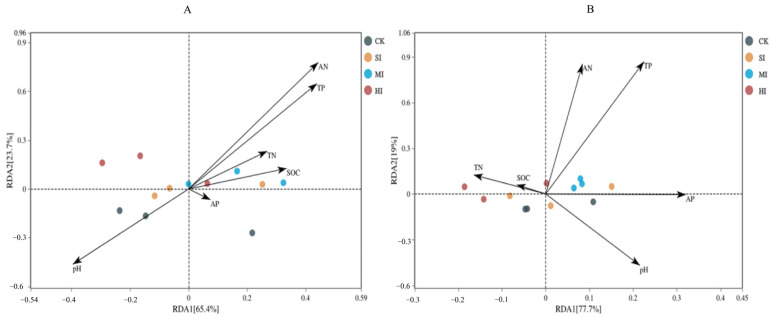
Redundancy analysis (RDA) of dominant soil microbial phyla (**A**) and genera (**B**) constrained by the analyzed soil physicochemical properties. CK, no expansion; SI, mild expansion; MI, moderate expansion; and HI, severe expansion.

**Figure 6 microorganisms-12-01648-f006:**
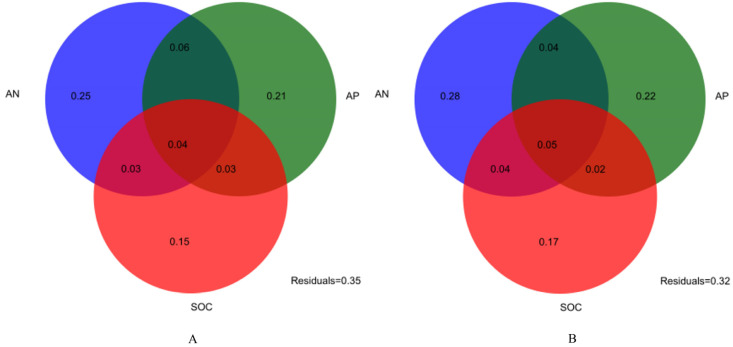
Variation partitioning analysis (VPA) of the relative contributions of AN, AP, and SOC to the soil microbial composition’s phyla (**A**) and genera (**B**). CK, no expansion; SI, mild expansion; MI, moderate expansion; and HI, severe expansion.

**Table 1 microorganisms-12-01648-t001:** Soil physicochemical properties with different N additions.

N Level	pH	SOC (g/kg)	TN (g/kg)	AN (g/kg)	TP (g/kg)	AP (mg/kg)
CK	5.35 ± 0.11 a	33.48 ± 3.87 a	3.01 ± 0.24 a	0.25 ± 0.02 b	0.84 ± 0.05 b	8.75 ± 0.61 b
SI	5.22 ± 0.11 a	39.85 ± 5.91 a	3.75 ± 0.26 a	0.31 ± 0.01 a	1.03 ± 0.07 a	8.25 ± 0.57 b
MI	5.26 ± 0.14 a	35.42 ± 4.41 a	3.27 ± 0.15 a	0.32 ± 0.02 a	1.07 ± 0.08 a	8.17 ± 0.92 b
HI	5.23 ± 0.12 a	34.44 ± 3.59 a	3.31 ± 0.21 a	0.34 ± 0.02 a	1.09 ± 0.09 a	10.12 ± 0.88 a

Note: All the results are reported as the mean ± standard deviation (*n* = 3), and the different letters within the columns indicate significant differences among treatments in the same season (*p* < 0.05). CK, no expansion; SI, mild expansion; MI, moderate expansion; and HI, severe expansion.

## Data Availability

Data are contained within this article.

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
