# Peer review of "Microbial C and N Metabolism Alterations Based on Soil Metagenome and Different Shrub Invasion Stages in Sanjiang Plain Wetlands"

_microorganisms, 2024, doi:10.3390/microorganisms12081648_

Round 1

Reviewer 1 Report

Comments and Suggestions for Authors

In the paper, the authors examined the effects of shrub expansion on microbial community composition and carbon and nitrogen cycling in wetland soils. The authors used several methods to determine the physicochemical properties of wetland soils, as well as metagenomic sequencing to determine the structure of microbial communities and genes involved in C and N cycling. The study found that the spread of shrubs significantly changed soil microbial composition and metabolic processes. function but had no significant effect on α-diversity.

The Manuscript is well written, but there are a few questions and comments:

1. An extremely short list of references; a small number of articles were used in the discussion; it is recommended to expand the chapter a little.

2. Lines 18-19: it may be worth removing the abbreviations from abstract, since only the abbreviation is used for control. And the control is further deciphered in the abstract text.

3. Section 2.1 contains incorrect references to the literature.

4. Section 2.4 does not contain any description of the methods - the determination of the physical and chemical properties of the soil should be briefly described

5. Section 2.5 – what database was used to determine the structure of microbial communities?

6. Information on calculating microbial community diversity indices should be added to section 2.6

7. Section 2.6 – what environment was the statistical analysis conducted in? Using which packages? What are the package versions?

8. Figures 2-4 are difficult to read. In particular, in Figure 4 the width of the columns should be reduced and the legend should be moved to the side of them.

9. Why was only the Shannon index determined? Other metrics are also used to determine α-diversity, for example, the Simpson and Chao1 indices.

10. Lines 24-25: replace the arrows with a verbal description.

11. Lines 163-170, 171-179: percentages of the content are given next to the names of types/genus - was the content the same for all samples or is it an average content? Should be clarified in the text.

12. Line 222 - replace ‘;’ with ‘,’ after SI

13. Line 223 - replace ‘.’ with ‘,’ before HI

14. Line 312, abstract: Why is it stated that nitrogen content is the main factor influencing the structure of microbial communities when it was previously also stated that soil pH, AN and TP are the main factors?

15. Also in the abstract it is stated that this study provides a deep understanding of the biogeochemical cycles of wetland ecosystems under conditions of shrub invasion (lines 25-26), and in the conclusion - This study still can not explain clearly that short-term shrub expansion was not sufficient to explain microbial carbon and nitrogen metabolism under shrub expansion (lines 316-318). Are there any contradictions between abstract and conclusion?

Best regards, reviewer.

Reviewer 2 Report

Comments and Suggestions for Authors

Overall, the manuscript is interesting and the topics covered quite relevant to considerations related to wetland ecosystems and biogeochemistry of essential elements (C, N). References to recent literature are also a strength of the manuscript. However, I also note shortcomings, which mainly concern the research methodology and presentation of results.The most important general comments:

1.       In my opinion, the main objection to the presented work is that on the basis of the presented methodology, I am not able to assess whether it was sufficient to enable the work to achieve its goal of determining changes in the metabolic pathways of the studied soil microflora.

- first of all, the entire chapter in the methodology on sequencing (2.5. DNA extraction, metagenome sequencing, and data processing) is almost identical to another paper (Yang, J., He, J., Jia, L., & Gu, H. (2023). Integrating metagenomics and metabolomics to study the response of microbiota in black soil degradation. Science of the Total Environment, 899, 165486). Nevertheless, the authors did not cite that paper.

- second, whether the obtained sequences have been entered into some database. It is necessary to state in the manuscript access to the raw data deposited in a database such as NCBI.

2.       Lines 110 – 111 - Please indicate In how many replicates was DNA isolation performer?

3.       Please note the numerous editorial and linguistic errors throughout the manuscript, such as: Line 112: „con- centration”, Line 124: „represen- tative”.

Reviewer 3 Report

Comments and Suggestions for Authors

The manuscript "microorganisms-3132296" should be reconsidered for publication due to minimal errors. The objective of the study is to analyse the impact on microbial diversity and activity of different levels of shrub invasion in a plain wetland of Chinese Sanjiang. The following are some aspects that I consider necessary to improve the quality and clarity of the manuscript: Abstract: It fulfils its objective, however, the conclusion reached by the authors of the study should be improved. Introduction: This section complies to some extent with its objective, some spelling errors should be corrected, as well as the establishment of references, however, it is necessary to establish the research hypothesis. Methodology: There is a sufficient description of the methodology; however, it is not clear about some statistical tests and how they were carried out. Results and discussion: In general, the figures of the results should be improved, since it is difficult to analyse them due to their size. The discussion should be expanded a bit more and focus on the ecological functions and their effect on the ecosystem. Conclusions: The results apparently support the conclusions of the study, however, to a large extent it should answer the research hypothesis, but as this is not established, there is no adequate closure of this section. The specific comments were placed in the pdf file, in the yellow highlights.

Comments on the Quality of English Language

NONE
